

**Earthquakes triggered by the subsurface undrained response to reservoir-**
**impoundment at Irapé, Brazil**
Haris Raza[1,2,3]*, George Sand França[1,2], Eveline Sayão[1], Victor Vilarrasa[3]
[1]Seismological Observatory, Graduate Program in Geology, Institute of Geosciences, University of
Brasília, Campus Darcy Ribeiro, 70297-400 Brasília, Brazil
[2]Institute of Astronomy, Geophysics and Atmospheric Sciences, University of São Paulo, 05508-090,
São Paulo, Brazil
[3]Global Change Research Group (GCRG), IMEDEA, CSIC-UIB, 07190 Esporles, Spain
*Correspondence to: Haris Raza (harisraza90@yahoo.com), Victor Vilarrasa (victor.vilarrasa@csic.es)
**Abstract**
The necessity to reduce carbon emissions to mitigate climate change is accelerating the transition from
fossil fuels to renewable energy sources. Specifically, hydropower, in particular, has emerged as a
prominent and safe renewable energy source, but entails reservoir-triggered seismicity (RTS). This
phenomenon causes significant challenges for safe reservoir management. Irapé, in Brazil, is a
prominent RTS site where seismicity surged after reservoir filling, with a maximum event of magnitude
3.0 in May 2006, just six months after the start of reservoir impoundment. Despite more than a decade
has passed since the seismicity occurred, the factors governing these earthquakes and their connection
to subsurface rock properties remain poorly understood. Here, we attempt to understand the potential
causes of RTS at Irapé dam, which is the highest dam in Brazil with 208 m, and the second highest in
South America. Permeability and porosity measurements of cylindrical cores from hard and intact rock
samples, which have been extracted near the RTS zone, by pitting 10 cm from the surface, reveal a low-
permeability rock. Porosity values range from 6.340 to 14.734%. Only 3 out of the 11 tested samples
present permeability higher than the lowest measurable value of the apparatus (0.002 mD), with the
highest permeability being 0.0098 mD. The undrained response of the low-permeability rock placed
below the reservoir results in an instantaneous increase in pore pressure and poroelastic stress changes
due to elastic compression, which brings potential faults located below the reservoir closer to failure
conditions. According to our analytical calculations, the increase in 136 m of the reservoir-water level
caused a 0.54 MPa pore pressure buildup at the depth of the Magnitude 3.0 earthquake, i.e., 3.88 km,
resulting in an increase of 0.82 MPa in the vertical effective stress and a decrease of 0.34 MPa in the





horizontal effective stress. These changes resulted in an increase in the deviatoric stress that led to fault
destabilization, causing the RTS. The laboratory measurements and analytical calculations corroborate
the hypothesis that the initial seismic activity was induced by the undrained subsurface response to the
reservoir loading at Irapé.
**Keywords**: Brazil, Reservoir-triggered seismicity, Permeability, Porosity, Fault, Reservoir-
management
**1.Introduction**
Reservoir impoundment, deep underground mining, and fluid injection into and withdrawal from the
subsurface are some of the well-known causes of induced/triggered seismicity which have become a
global issue in the past few decades (McGarr et al., 2002; Foulger et al., 2018; Kivi et al., 2023). The
understanding and identification of these types of human-induced earthquakes is crucial in terms of
environmental and economic impact, as well as for socio-political and scientific discussion (Gonzalez
et al., 2012; Vilarrasa et al., 2019). Recently, the debate over potential induced or triggered nature of
cases of felt seismicity has intensified, such as the Oklahoma earthquakes of Mw 5.7 in 2011 and of
Mw 5.8 in 2016 (Ellsworth, 2013; Keranen et al., 2013; Yeck et al., 2017), Emilia, Italy, earthquakes of
Mw 6.1 and 5.9 in 2012 (Cesca et al., 2013a), Pohang, South Korea, earthquake of Mw 5.5 in 2017
(Grigoli et al., 2018; Kim et al., 2018), Lorca, Spain, earthquake of Mw 5.1 in 2011 (González et al.,
2012), and Castor, Spain, earthquake sequence of  Mw 4.1 in 2013 (Cesca et al., 2014; Vilarrasa et al.,
2021; Vilarrasa et al., 2022 ), to name a few. Apart from the possibility of injuring people and damaging
infrastructure, such earthquakes can have a negative public perception leading to project cancellation
(Boyet et al., 2023a).
The first reservoir-triggered seismicity (RTS) case was observed during the filling of Lake Mead at
the Hoover Reservoir (US) in the mid-1930s, with ~M4.0 (Carder 1945). Major worldwide RTS cases
were detected in the 1960s, such as the M6.1 Hsinenghiang (China) in 1962, Kariba (Zambia) M6.2 in
1963, Kremasta (Greece) M6.3 in 1966, and Koyna (India) M6.3 in 1967 (Gupta, 2002). To date, over
150 RTS cases have been documented (Wilson et al., 2017; Foulger et al., 2018). Studies to understand



the triggering mechanisms of RTS show that pore pressure changes in the order of a few tenth of MPa
and the associated poroelastic stress changes are sufficient to reactivate deep faults (Rice and Cleary,
1976; Simpson, 1976; Bell and Nur, 1978; Talwani and Acree, 1985; Roeloffs, 1988; Simpson et al.,

1988).

RTS is generally controlled by the stress state, the geological and hydrogeological properties of the

region, and the water-level changes at the reservoir. The perturbation caused by the changes in water-
level results in the loading and/or unloading of the subsurface, which may respond in an undrained or
drained way. An undrained response leads to an instantaneous pore pressure buildup that is proportional
to the height of the reservoir load. In contrast, a drained response leads to pore pressure diffusion into
the rock that causes progressive pore pressure build-up as the pressure front propagates into the rock
(Table 1). In general, RTS magnitudes are smaller for undrained responses than drained ones (Simpson
et al.,1988). The interactions and comprehensive analysis of these two responses are key to improving
the forecasting and mitigation of RTS hazard.
**Table 1.** The time-distribution types of responses to reservoir-triggered earthquakes (by Simpson, 1988)

| Response type | Mechanism | Description | Main features | Cases |
|---|---|---|---|---|
| **Instantaneous response** | Instantaneous elastic response and undrained response due to reservoir loading | This type of RTS increases immediately after the initial impoundment of reservoir or changes rapidly after rapid changes in the water level. | Changes in water level have a strong correlation with the change of seismicity, this generally occurs around the reservoir area, and the earthquake magnitude is small, the majority of them are swarm seismicity. | Monticello, Manico-3, Nurek, Kariba, Kremesta *Irapé (this paper)* |
| **Delayed response** | Increase of pore pressure caused by pressure diffusion through permeable rock below the reservoir | It is only after a period of reservoir impoundment that the seismicity changes continuously | No significant correlation between changes in water level and seismicity, the time delay is obvious, the magnitude is generally large, and the earthquake occurrence point is not limited. | Koyna, Aswan, Oroville |


The RTS cases are booming around the world, with Brazil being one of the concerned countries with
29 RTS cases to date (Sayão et al., 2020). The study of RTS in Brazil started in 1972 with the M3.7 at



Carmo do Cajuru reservoir, southeast Brazil (Foulger et al., 2018). The largest recorded event, a M4.2
in 1974, caused damage to several buildings without any fatalities and was associated with nearby
reservoirs at Porto Colombia and Volta Grande, both of which started damming in the early 1970s
(Sayão et al., 2020).
The Irapé dam, located in the state of Minas Gerais, Brazil, is the highest dam in Brazil with about
208 m, and the second highest in South America (França et al., 2010). The Irapé hydropower plant lies
in the vicinity of Jequitinhonha River. Seismicity started to increase immediately after the impoundment
of the reservoir and completion of the dam with the maximum event of M3.0 occurred on 14 May 2006,
coinciding with the peak water level of the dam. The significant magnitude of the earthquake and the
early occurrence after-filling of the reservoir impoundment has raised questions about the triggering
mechanisms of this RTS. Understanding these mechanisms is crucial for ensuring the safety of
infrastructure around the Irapé reservoir and for the local population.
In this study, we aim to investigate the potential causes of the main RTS event at Irapé. We initially
elaborate on the geological setting and rock characteristics in the vicinity of the reservoir. We explain
the characteristics of the RTS at Irapé, including the temporal evolution of the seismicity, which
occurred in the short period from December 2005 to May 2006 and the location of the main event based
on the local velocity model. Then, we present the performed permeability and porosity tests of
cylindrical cores from hard and intact rock samples, which have been extracted near the RTS zone to
identify and describe the primary role of porosity and permeability. We perform analytical calculations
to estimate that pore pressure and poroelastic stresses in response to the highest water level of the
reservoir filling and the time that would take for the pore pressure diffusion front to reach the depth of
the main event. We present evidence that the cause of RTS at Irapé was the undrained response of the
subsurface to reservoir impoundment.
**2. Geological setting and RTS at Irapé**
**2.1 Geological setting**



The area of Irapé is located within the domain of the Pre-Folding Belt Cambrian Araçuaí, which is
oriented approximately in a north-south direction and defines the eastern part of the São Francisco
Craton in the State of Minas Gerais (Almeida, 1977). Approximately 80% of the reservoir area in Irapé
corresponds to the Chapada Acauã Formation. The Chapada Acauã Formation, which has been
investigated near the Irapé Shear Zone (Araujo et al., 2010), consists of carbonaceous mica-schist rocks,
locally with pyrite, garnet, or graphite (Lima, 2002). This rock is intensely deformed, with the formation
and rotation of quartz sub-grains and migration of grain edges. It represents, together with the Nova
Aurora Formation, typical sedimentation of passive margin associated with deposition in the Macaúbas
Basin. To the east of the Chapada Acauã Formation, it is found the Ribeirão da Folha Formation,
consisting of mica shales, metaritmitos, quartzite and calc-silicates rock (Figure 1).



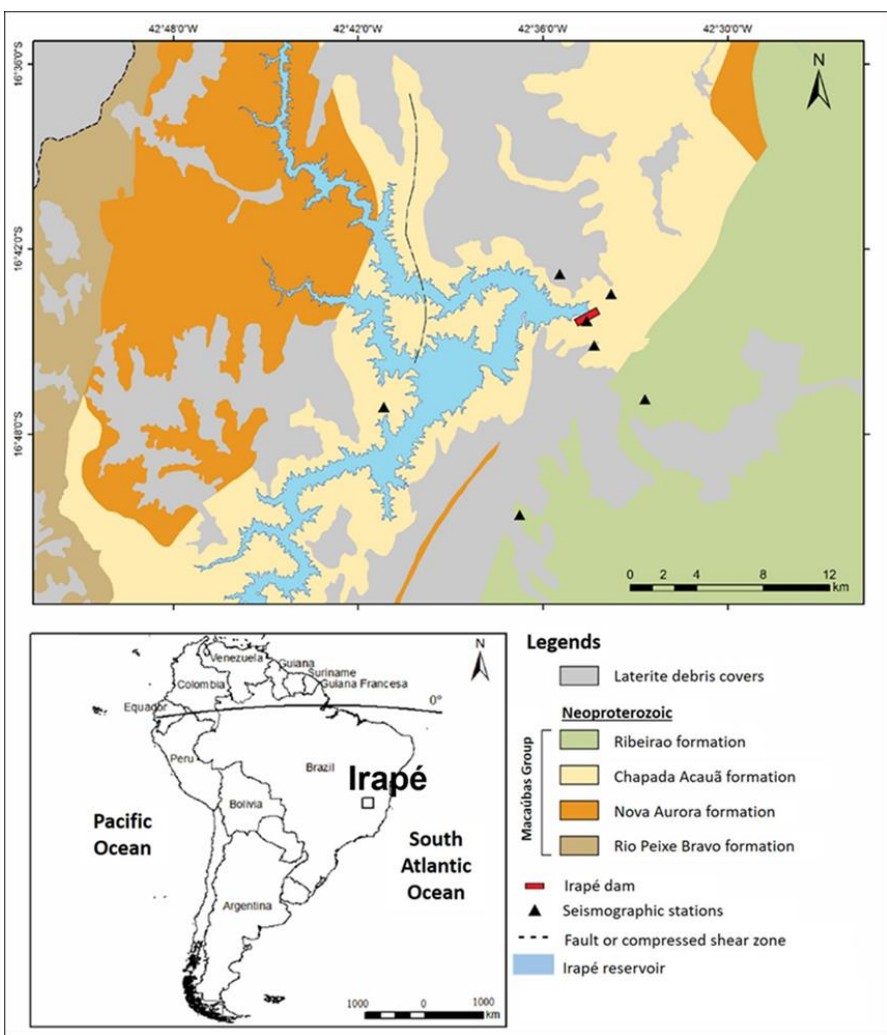
**Figure 1.** Geological map of Irapé reservoir and surrounding area
**2.2 Background on the reservoir-triggered seismicity at Irapé**
The Irapé reservoir covers an area of 137.8 km$^2$ with a reservoir volume of 5.964 km$^3$. The dam was
constructed on the Jequitinhonha River, filling the reservoir to a maximum height of 137 meters (Figure
1 and Table 2). The dam area was monitored by a three-component seismic network at three stations
prior to 3 years of its impoundment, which started on 7 December 2005. These stations did not detect
any seismicity before the impoundment (Chimpliganond et al., 2007).



**Table 2.** Characteristics of the main RTS event at Irapé (França et al., 2010)

| Dam height (m) | Length (m) | Volume (km³) | Max. reservoir water depth (m) | Reservoir area (km²) | Seismicity type | Date | Magnitude (mR) | Io (MMI) | ΔT(yr) |
|---|---|---|---|---|---|---|---|---|---|
| 208 | 540 | 5.964 | 137 | 137.8 | Initial | 14 May 2006 | 3.0 | III-IV | 0.5 |

*ΔT: interval time (years) since the start of filling/impoundment; MMI: modified Mercalli*
*Intensity scale, mR: magnitude Regional.*
Microearthquakes started to be detected just one day after the impoundment began, exceeding 300
microearthquakes by October 2006. The largest event occurred on 14 May 2006 with a M3.0 that was
felt at the reservoir area (Chimpliganond et al., 2007; França et al., 2010). The seismicity occurred
within a small area, with epicentres in the lake and its nearby margins (less than 3 km from the narrow
lake), close to the dam axis. The evident time correlation between the start of the impoundment of the
lake and the occurrence of seismicity suggests a causative relationship for this seismicity (Figures 2 and
3). The spatial distribution of the epicentres also suggests the hypothesis that this is another case of RTS
of the initial response type.

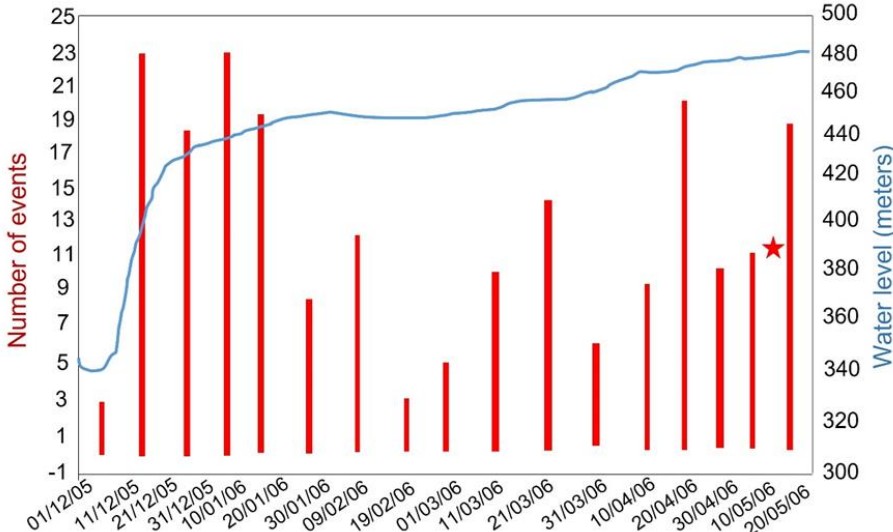


**Figure 2**. Temporal evolution of RTS at Irapé by ten days. Number of events during December 2005 to
May 2006 (histogram) at Irapé and average water elevation above the mean sea level (blue line) are



illustrated. The red star indicates the time when the main and largest event occurred, M3.0 on 14 May
2006 (modified from Silva et al., 2014).

The events were analysed using the program Seismic Analysis Code (Goldstein and Snoke,2005), in

which the arrival of the P and S waves and the polarity are considered. The hypocentre location of the
events that were recorded by three stations was computed with the program HYPO71 (Lee and Lahr,
1975). The analysis of seismograms went through a double-checks routine (Silva et al., 2014).

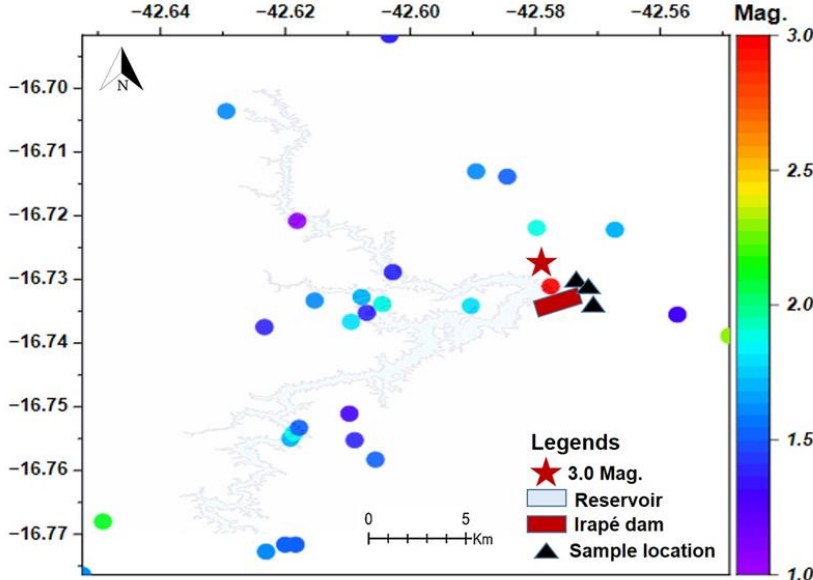


**Figure 3.** RTS Distribution in the initial period with location and magnitude (see colour scale), the red
star is the main event felt near the dam and black triangles denote the samples location.

Velocity models were adopted based on a deep seismic refraction survey in combination with local

geological interpretations and studies of the crustal structure in south-eastern Brazil to locate seismic
events in the Irapé area (Assumpção et al., 2012). The local velocity model consists of a superficial 4.8
km-thick layer with a P-wave velocity ($V_p$) of 4.5 km/s, representing the mica-schist to graphite-schist
rocks from surface, and a second layer from schist to crystalline basement rocks with a thickness of
11.2 km with P-wave velocity ($V_p$) of 6 km/s (Marshak et al., 2006; Silva et al., 2014).





The repetition of a structural trend in the NE-SW direction originates from the geological and
geophysical structuring of the crust (Silva et al., 2014). The stress regime in the Irapé region has been
estimated to be a normal faulting stress regime. The accuracy of the focal mechanisms remains a subject
of debate due to the low quality of the seismic data recorded by analogue seismograms and uncertainties
associated with the velocity model. Consequently, the focal mechanisms of the May 14, 2006, M3.0
earthquake have not been resolved yet (Silva et al., 2014).
**3.Materials and methodology**
We inspected the Irapé site and surrounding areas as well as the outcrops. The dam area is surrounded
by mica-schist rock, which is shiny, ranging from blackish to medium grey in colour, with foliated, fine
to medium-grained textures. According to the local velocity model, there is a superficial layer that is
4.8-km thick, representing mica-schist to graphite-schist rocks at the surface. Below that, there is a
second layer that is 11.2-km thick, consisting of crystalline basement rock. Measurements from these
samples are crucial for understanding the estimated permeability beneath the subsurface in the context
of the main event, which occurred at a depth of 3.88 km (França et al., 2010). Since the epicenter of the
main event was located about 1 km away from the dam, we collected bulk rock samples from different
locations around the dam, as well as nearby outcrops, by digging pits that were 0.10-m deep.
**3.1 Laboratory experiments**
We have extracted cylindrical core samples perpendicular to the bedding planes of mica-schist rock.
We have performed tests on three sets of samples, with a total of 11 core samples, of hard and intact
samples because the rest of the samples were fragile and fractured during the coring from bulk samples
(Table 3). The retrieved cylindrical plugs have a length ranging from 3.8 to 5.0 cm and a diameter of
2.50 cm, which meets the International standard criteria (Core Lab) to measure core plug samples by
Ultra-Pore 300 and Ultra-Perm 610 (Figures 4).



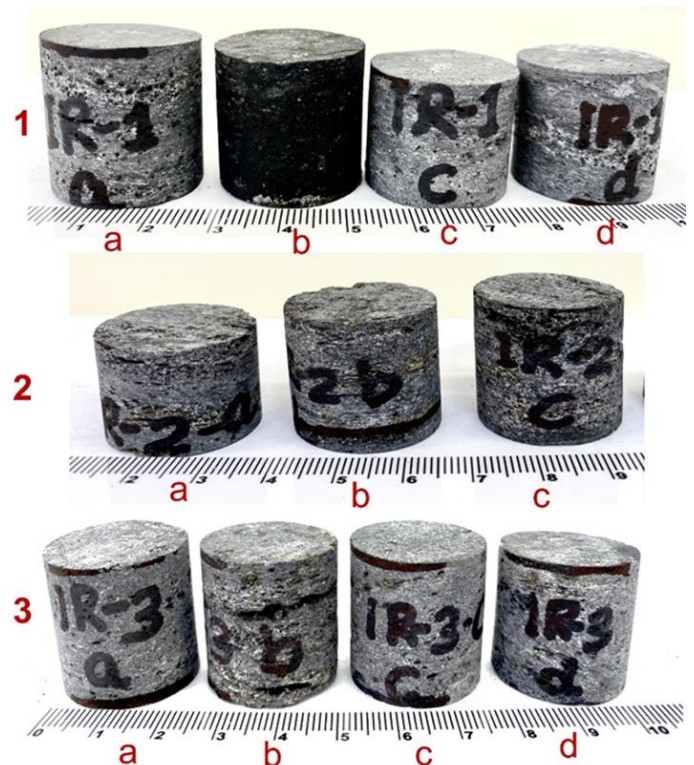

**Figure 4.** The three sets of mica-schist rock samples (1, 2, and 3) after cylindrical coring from bulk samples (⊥ coring of cylindrical plugs has been done by loading perpendicular to the bedding planes).

We conduct porosity measurements using the Ultra-Pore 300, which is manufactured by Core Lab Instruments in Texas, USA. The Ultra-Pore 300 is a gas expansion helium pycnometer specifically designed for determining the grain volume or pore volume of both core plug and full-diameter samples. To achieve this, we utilized matrix cups designed for samples with diameters ranging from 2.5 to 3.8 cm, equipped with a Setra 204 transducer rated for pressures ranging from 0 to 1.72 MPa. We determined the pore volume using the nitrogen gas ($N_2$) expansion technique (API,1998; Ceia et al., 2019).

We measure the intrinsic permeability of rock samples using Ultra-Perm 610 Permeameter. This precision equipment, which controls backpressure, maintains a constant rate or mean pressure at 0.69 MPa. Before testing, samples are cleaned with soxhlet equipment and toluene, followed by drying in an



182 oven. The permeability measurements included a permeameter, nitrogen source, stopwatch, a core

183 holder, a bubble tube, and a digital calliper. The core holder is pressurized to 3.45 MPa confining

184 pressure using compressed air. The bubbles passing through a burette are timed, and outflow gas volume

185 is recorded. The permeability is calculated using Darcy's law, considering core dimensions. Hard rock

186 core samples, like mica-schist, require long stabilization times due to the low permeability.

187 **3.2 Analytical calculations of undrained pore pressure and stress changes**

188 Reservoir impoundment causes an undrained effect in the subsurface that manifests as an instantaneous

189 pore pressure and stress changes below the reservoir (Skempton, 1954). The change in the vertical

190 stress, $\Delta\sigma_v$, equals the weight of the water level rise. The horizontal stress, assuming iodometric con-

191 ditions, changes proportionally to pore pressure changes as (Rutqvist, 2012)

$$\Delta\sigma_h = \alpha \frac{(1-2\nu)}{(1-\nu)}\Delta p \,, \tag{1}$$

193 where $\Delta\sigma_h$ is the horizontal stress change, $\alpha$ is Biot's coefficient, $\nu$ is Poisson's ratio and $\Delta p$ is the pore

194 pressure change. Additionally, in an isotropic and homogeneous poroelastic material subject to un-

195 drained conditions, the change in pore pressure resulting from a change in stress can be computed as

196 (e.g., Rice and Cleary, 1976; Cocco and Rice, 2002)

$$\Delta p = \frac{-B}{3}\Delta\sigma_{kk} \,, \tag{2}$$

198 where $\Delta\sigma_{kk} = \Delta\sigma_v + 2\Delta\sigma_h$ , $\Delta\sigma_{kk}$ is the stress change and $B$ is the Skempton's coefficient of mica-

199 schist rock (Roeloffs, 1988). Equations (1) and (2) constitute a system of two equations with two un-

200 knowns. Its resolution yields the undrained pore pressure change as

$$\Delta p = \frac{B}{3}\frac{\Delta\sigma_v}{\left(1 - \frac{2B\alpha(1-2\nu)}{3}\frac{(1-2\nu)}{(1-\nu)}\right)}. \tag{3}$$

202 **3.3 Analytical calculations of the time at which the pore pressure diffusion front reaches the**

203 **depth of the earthquake**

204 The advancement of the pore pressure front within the subsurface is controlled by diffusivity



$$D = \frac{k\rho g}{\mu S_s}$$
(4)

where, $D$ is diffusivity, $k$ is the intrinsic permeability, $\rho$ is water density, $g$ is gravity, $\mu$ is water viscosity,
and $S_s$ is the specific storage coefficient. The time at which the pore pressure front reaches a certain
distance $r$ is given by
$$t = \frac{r^2}{D}$$
(5)

**4.Results**
**4.1 Porosity and permeability measurements**
The results of our laboratory measurements are provided in Table 3. These data are subject to meas-
urement uncertainties inherent to the experimental equipment used according to the standard procedure.
Laboratory measurements of samples of mica-schist reveal a low permeability (Table 3 and Figure 6).
The maximum permeability is 0.0098 mD, but most of the samples present a permeability below the
precision of the apparatus, i.e., lower than 0.002 mD. Such permeability is in the range of low-permea-
bility rock, which act as a barrier to flow. Most of the samples have a porosity between 6 to 10%, except
for two with higher porosity. The low permeability of mica-schist could be explained by the fact that
the larger pores are not well connected (Figure 5). In general, there is no correlation between permea-
bility and porosity (Figure 6).

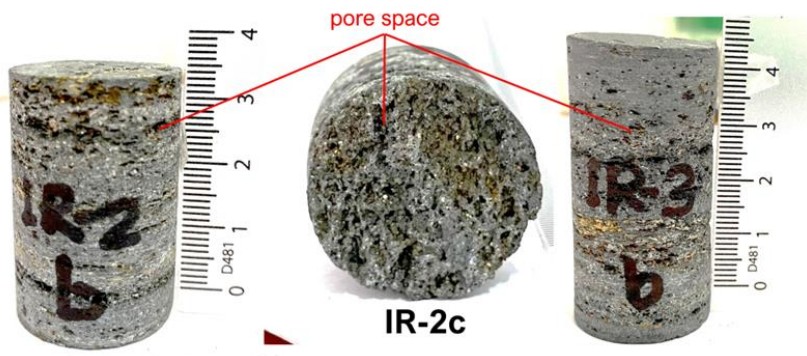






**Figure 5.** Megascopic representation of samples IR-2 b, c, and IR-3b showing pores that are not well-connected.

**Table 3.** Location of samples with permeability and porosity data from measured cores

| Location (lat., long.) | Sample Numbers | Permeability (mD) | Porosity (%) |
|---|---|---|---|
| 16.73872, 42.57680 | IR-1a | 0.002 | 7.529 |
| | IR-1b | 0.002 | 6.785 |
| | IR-1c | 0.002 | 8.781 |
| | IR-1d | 0.0098 | 6.555 |
| 16.74038, 42.57652 | IR-2a | 0.002 | 9.490 |
| | IR-2b | 0.0038 | 10.465 |
| | IR-2c | 0.0038 | 14.734 |
| 16.72438, 42.56316 | IR-3a | 0.002 | 6.943 |
| | IR-3b | 0.002 | 13.323 |
| | IR-3c | 0.002 | 7.126 |
| | IR-3d | 0.002 | 6.340 |

*Experiments loaded perpendicular to bedding plane* ($\perp$)

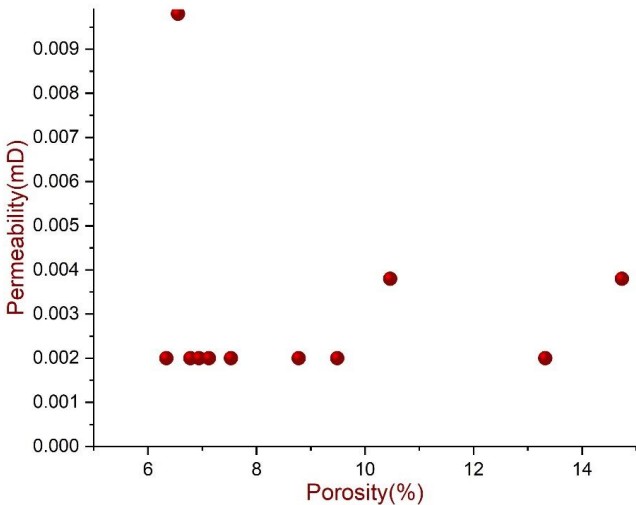

**Figure 6.** Porosity-permeability relation of mica-schist rock samples.

**4.2 Undrained response of rock: changes in pore pressure and stress**

The 136 m of water level increase at the time of the M3.0 earthquake resulted in an increase in the

vertical stress of 1.36 MPa. To compute the pore pressure change caused by the reservoir impoundment,

the Biot coefficient, Skempton's B coefficient and Poisson's ratio of mica-schist are needed (Eq. (3)).

Since such measurements are not available, we adopt the values of Opalinus Clay because it is a similar



rock to mica-schist. Thus, we assume Skempton's B coefficient of 0.92, undrained Poisson's ratio of
0.39 and Biot's coefficient of 1. With these values, the resulting pore pressure change is 0.54 MPa.
Consequently, the horizontal stress change is of 0.19 MPa (Eq. (1)). These pore pressure and stress
changes result in a vertical effective stress increase of 0.82 MPa and a horizontal effective stress
decrease of 0.34 MPa, increasing the deviatoric stress in more than 1 MPa.

### 238    4.3 Pressure diffusion along mica-schist

The measured intrinsic permeability of mica-schist is in the order of $10^{-18}$ m$^2$ (Table 3). Assuming a
specific storage coefficient in the order of $1.05 \times 10^{-6}$ m$^{-1}$, diffusivity (Eq. (4)) results in $9.5 \times 10^{-6}$ m$^2$/s.
Taking into account that the depth of the M3.0 earthquake occurred at 3.8 km, the time at which the
pore pressure front would reach this depth by diffusion (Eq. (5)) is in the order of 50,000 years.

### 243    5.Discussion

RTS has been the focus of many studies, but the origin and development of RTS are still unclear in
many cases (Gupta et al., 2016; Arora et al., 2018). There is a general consensus that there are two main
triggering mechanisms (Simpson et al., 1988). On the one hand, low-permeability rock has an undrained
response to the water-level changes of the reservoir, which acts as a loading, instantaneously increasing
pore pressure and causing poroelastic stress changes deep underground (Chen and Talwani, 2001;
Vilarrasa et al., 2022; Raza et al., 2023). On the other hand, in the presence of permeable rock or a
permeable fracture network, pore pressure diffuses downwards, which may eventually trigger an
earthquake if a critically stressed fault becomes pressurized (Talwani and Acree, 1985).
At Irapé, the low-permeability of the rock below the reservoir, i.e., mica-schist with permeability in
the order of $10^{-18}$ m$^2$ or lower, hinders pore pressure diffusion. Given that the hypocentre was located at
3.88 km depth, the pressure propagation front would take in the order of 50,000 years to start
pressurizing the depth at which the earthquake was nucleated. Even assuming that the presence of
fractures enhanced the rock permeability by three orders of magnitude, which would be the upper limit
of observed permeability enhancement of low-permeability rock at the field scale (Neuzil, 1986), the
pressure front would take 50 years to reach 3.88 km depth. The necessary permeability of the rock to



reach the depth of the largest earthquake within 0.5 years, i.e., the delay of the earthquake with respect
to the start of impoundment, would be of $10^{-13}$ m$^2$, five orders of magnitude higher than the actual
permeability of mica-schist. Such high permeability enhancement is deemed unlikely.
Considering the load caused by the water-level rise in the reservoir of 136 m, the low-permeability
mica-schist experienced an undrained response, with subsequent poroelastic stress and pore water
changes. We have estimated these changes analytically, finding a vertical effective stress increase of
0.82 MPa, a horizontal effective stress decrease of 0.34 MPa, and a pore pressure increase of 0.54 MPa.
Given the normal faulting stress regime at Irapé, these changes cause an increase in the deviatoric stress
that could destabilize faults in the subsurface. These changes in pore pressure and stress levels provide
valuable insights into the dynamic behaviour of the geological formation and are crucial considerations
in understanding the reservoir response to alterations in reservoir water levels. We contend that the rapid
loading of the reservoir weakens this fault because of the undrained stress and pore pressure changes
(Figure 7).
In addition, the megascopic representation of core samples in the configuration of the physical
evidence illustrates that rock can exhibit relatively high porosities and low permeability when their
pores are not well-connected (Figure 5). Thus, mica-schist may present preferential lateral fluid
migration at depth, following the foliation direction. The surface rock beneath the Irapé reservoir is
highly metamorphosed and generally has good porosity and low permeability. Therefore, pore pressure
diffusion is disregarded as the potential cause triggering the seismicity at Irapé.



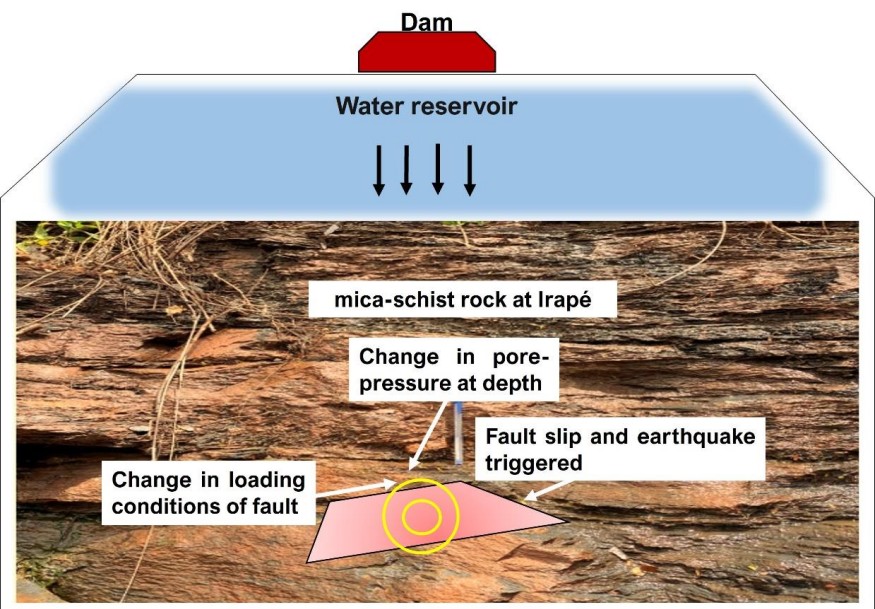

**Figure 7.** Schematic description of the mechanism of RTS at Irapé, indicating the effect of the weight of the reservoir water volume due to undrained response in low-permeable mica-schist rock (the background photo was taken in the field from an outcrop at Irapé).

The regional geology at the eastern part of the São Francisco Craton in the State of Minas Gerais follows a N-S direction (Almeida, 1977). Silva et al. (2014) also mentioned that the repetition of a structural trend in the NE-SW direction originates from the geological and geophysical structuring of the crust. This trend makes it feasible to assume the existence of a N-S vertical mature fault that could become destabilized by small changes in the effective stress. An association of such seismicity with the shear zone along reservoir /lineaments suggests the reactivation of such faults under the influence of reservoir impoundment.

To mitigate the risk of RTS, it is crucial to thoroughly characterize the site by measuring rock physical properties. Analytical and numerical solutions should integrate the physics of the problem, such as poromechanics to assess both the undrained response of the subsurface to reservoir impoundment and pore-pressure diffusion. Such models should include the rock layers below the reservoir down to the crystalline basement and their characteristics, including features like faults. Before the construction of



the dam, the hazard of triggering moderate to large earthquakes should be estimated, to disregard
locations with high probability of RTS. This estimation requires knowing the hydro-mechanical
properties of the rock layers, i.e., permeability, porosity, stiffness, and strength, as well as the design
parameters of the dam, i.e., height. The successful management of RTS requires an interdisciplinary
approach combining concepts of hydrogeology, geomechanics and seismology.
Finally, to address and manage RTS risks, the traffic light protocol (TLP) is being employed. In general,
TLP initiates the green light as the primary approach allowing operations without restrictions, the yel-
low light as the point to activate mitigation measures, and the red light as the point necessitating regu-
latory intervention. The efforts have also begun by linking the configuration of TLPs with risk-oriented
measures, infrastructure harm, and the likelihood of loss or damages while adapting them to real-time
data. The occurrences that may ensure after an operation are crucial, given their substantial impact on
standard risk management. Nevertheless, these methodologies can be revolved around by assessment
of events succeeding in the conclusion of an operation. The utilization of physics-based models holds
promising by illustrating and projecting anticipated seismic activity, enabling the anticipation of future
warnings and risks, and build up the information for operational adjustments and for future mitigation
(Boyet et al., 2023b) (Figure 8).





**Figure 8.** Reservoir operations and impoundment are strategically designed to reduce the risk of RTS. Monitoring seismic and geophysical activities yields information for predictive earthquake models. The catalogues of earthquakes and source/origin models are applicable in the assessment of hazard and risk. These assessments of risk and hazard can guide the development of a traffic light protocol (TLP), functioning as a dynamic decision module during operations. The display of each box shows the classifications of input data (blue boxes) and output results (grey boxes).

Regarding the mitigation approaches for RTS within the framework of a TLP, the effectiveness of an operator heavily relies on the efficiency of mitigation strategies implemented at the yellow-light stage. Ideally, these strategies would proficiently diminish seismic risks and hazards, ultimately circumventing the red-light scenario that terminates the operation. Thus, TLPs can be one major strategy and strong decision-making tool for operators to minimize the risk of RTS for future developments of dams.

**6.Conclusions**

We have analysed RTS at Irapé to discern the cause of the triggered seismicity. The measured low permeability of the rock at Irapé disregards pore pressure diffusion as the triggering mechanism and suggests that the M3.0 RTS was triggered by the undrained response of the subsurface to reservoir impoundment. Analytical calculations estimate that pore pressure increased by 0.54 MPa in response to an increase of 136 m in the reservoir-water level. The resulting vertical effective stress increased by 0.82 MPa and the horizontal effective stress decreased by 0.34 MPa. Thus, the deviatoric stress would increase in a normal faulting stress regime, like the one at Irapé, destabilizing the fault and causing RTS. Both laboratory measurements and analytical calculations support the hypothesis that the initial seismicity was triggered by the undrained response of the subsurface to the loading of the reservoir at Irapé. This study also suggests that the occurrence of such earthquakes may be avoided by carefully manipulating reservoir loading.

**Data availability**

The data analysed and /or used in this study are presented in the Supplementary Material.



**Supplementary Material**

The Supplementary Material related to this article is available online.

**Author contributions**

H.R., G.S.F., V.V. co-designed the study. E.S. and H.R. did sampling. H.R. wrote the paper performed laboratory measurements. H.R. and V.V. did the analytical calculations. G.S.F. and V.V. reviewed, contributed to the interpretation of the results, and edited the paper.

**Competing interests**

The corresponding and co-authors state that there are no competing interests.

**Acknowledgments**

This study was financed in part by the Coordenação de Aperfeiçoamento de Pessoal de Nível Superior -Brasil (CAPES) -Finance Code 001. The authors acknowledge funding from the Spanish National Research Council (CSIC) under the Program for Scientific Cooperation iCOOP+ through the Project COOPA20414. V.V. acknowledges funding from the European Research Council (ERC) under the European Union's Horizon 2020 Research and Innovation Program through the Starting Grant GEoREST([www.georest.eu](www.georest.eu)) under Grant agreement No. 801809. IMEDEA is an accredited "Maria de Maeztu excellence Unit" (Grant CEX2021-001198, funded by MCIN/AEI/ 10.13039/501100011033). G.S.F gratefully acknowledges CNPq (Grant 310240/2020-4 PQ-1C). G.S.F. and H.R. and E.S. also thank to INCTET-CNPq (Institutos Nacionais de Ciência e Tecnología de estudos tectónicos) Brazil. We thank to Prof. Carlos Jorge de Abreu for conducting samples measurements at Laboratory of Physical Properties of Rocks at the University of Brasília.

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
