# Peer review of "Earthquakes triggered by the subsurface undrained response to reservoir- impoundment at Irapé, Brazil"

_EGUsphere, 2024_

## Referee Comment (RC2)

The article presents an analytical assessment of the undrained response impact on the state of stress change during reservoir impoundment and associated reservoir triggered seismicity. Authors present laboratory measured permeability and porosity to support the assumption of undrained response. The assessment is based on assumption of oedometer conditions and using analytical expression based on poroelastic relationships. However, a significant deficiency is that Equation 1 has only the relation between horizontal stress and pore pressure. Even if the pore pressure is not changing (drained condition) the horizontal stress would change due to the change of vertical stress. This effect is expected to be the same order of magnitude or stronger than the effect of pore pressure change and therefore cannot be neglected. It is recommended that the calculations must be updated or it should be explicitly shown that the term can be neglected (more details are available in specific comments in attached file.

**Specific comments:**

12: Either "Specifically" or "in particular"

22: Is the accuracy of porosimetry sufficient to report it to third decimal places?

69: The delayed response is dominant in case of the Koyna, however the local maximum of seismic activity is also observed during the maximum water level (http://dx.doi.org/10.1134/S1069351322030077). It might be worthy to mention that both mechanisms are present, but the dominant one is delayed.

122-123: It would be great to mention the depth of earthquakes here.

127: The number of events cannot be negative, so it would be better to start the vertical axis from 0 and not -1.

134: Since only three stations were utilized for the location, it seems to be impossible to determine 4 parameters of earthquake hypocenter (x,y,z,t) without certain assumptions. It would be great to mention if assumptions were made and what is average accuracy of location.

184-186: It would be great to mention the lower instrumental limit for the permeability and the accuracy of measurements. It also seems to be inconsistent with the information in supplementary material, which says that 0.01 mD is the lower limit. In general, supplementary material contains important information but never referred in the main manuscript which make it difficult to find.

**192 MAJOR:** The equation consider only relationship between change in horizontal stress and change in pore pressure. However, the change in vertical stress would also cause change in the horizontal stress, which is neglected. The equation should be

$$\Delta\sigma_h = \frac{v}{1-v}\Delta\sigma_v + \alpha\frac{(1-2v)}{1-v}\Delta p$$

197: What is the sign agreement here? If the pore pressure is positive and compressive stresses are also positive, it should be no minus sign.

198: "*mean* stress change"

201: Equation seems to assume that there is no minus sign in Equation 2. **MAJOR**: The equation have to be modified due to the change in Equation 1. Or it should be explicitly shown why term proportional to change of vertical stress in neglected.

224: Is the porosimetry is accurate to report it to three decimal places? It is suggested that "<0.002" is reported instead of "0.002" for permeability.

226: Since the measurement are close to instrumental limit, it would be great to show this limit on the plot as well as the accuracy of calculated permeability.

242: It is mentioned further in the text, but it would be great place to say that absence of fractures is assumed.

259-261: The permeability measured in the lab is usually 1-2 orders of magnitude lower than the in-situ one even without visible fractures. At least in the laboratory experiments, the presence of open fractures is capable of enhancing the permeability by 5 orders of magnitude (e.g., https://doi.org/10.1038/s41598-022-19775-4 or https://doi.org/10.22541/essoar.171629597.77744897/v1). It is acceptable to assume that 5 orders of magnitude enhancement is unlikely, but sightly extended discussion would be worthy here.

334-335: It is not fully clear what is meant by "carefully manipulating reservoir loading". If the change of impoundment rate is meant, it seems unlikely that the impact of undrained response will be different by the end of impoundment. If the limiting the maximum water level is meant, it is not fully clear what is suggested as a limit and rewording might be necessary.

---

## Author Response (AR1)

**Response to reviewers comments on the paper "egusphere-2024-166"**

We discuss below the comments made by two reviewers, our response to them and how we address them in the manuscript. To facilitate reading, the original comments are provided in a standard font, our responses in italics blue font, and a summary of the changes made in the text in italics green font. Page and line numbers of the manuscript with track changes are used to address changes made in the text.

**Response to comment by Referee #1**

**1-514:** The manuscript discusses the impact of reservoir filling on induced seismicity and concludes, that the undrained response of the subsurface rather than pore pressure diffusion is triggering mechanism. Field, lab and modelling techniques are used to reach this conclusion. Recommendations on how to prevent future induced earthquakes through reservoir impoundment strategies are given.

The scientific significance of the manuscript is high with the topic remaining urgent and unresolved. The presentation quality is superb as the manuscript is very well structured and filled with visually pleasing and comprehensible figures. The use of the English language is impeccable throughout most parts; however, two sections fall below standard and need thorough revision. The list of references also needs revision as consistency is lacking. The scientific quality is high as well. Abundant and relevant references are included, established methods are applied, results are discussed and put into context, and recommendations for reservoir management strategies are given.

*Author's response: We extend our thanks to the reviewer for his/her careful consideration and positive assessment of the paper. We have addressed the comments of the reviewers, which have served to clarify some of our assumptions and, as a result, we feel that the quality of the manuscript has improved.*

13: At what frequency? It sounds like it is always the case.

*Author's response: Yes, seismicity is triggered in almost all cases. Generally, microearthquakes are observed soon after or during the reservoir impoundment, which corresponds to the initial undrained response. In some cases, larger earthquakes occur many years later, in what is so called delayed response. RTS cases with a magnitude M > 3.0 are typically a concern. In the early 1960s, several cases of reservoir-triggered seismicity (RTS) with magnitudes of 6 or more were recorded (lines 52-59).*

18: Great!

*Author's response: Thanks!*

21: Delete comma

*Author's response: Deleted.*

22: Numbers until twelve in letters?

*Author's response: We have corrected.*

23: Wording too complicated. Maybe "above the bottom threshold".

*Author's response: We have reworded a bit the sentence. Now it reads "Only three out of the eleven tested samples present permeability above the lowest measurable value of the apparatus".*

29: Traditionally I expect all effective stresses to decrease when pore pressure increases. Please make difference clearer in this sentence.

*Author's response: We appreciate the reviewer's insightful comments. Indeed, the coupled hydro-mechanical response of the subsurface to reservoir loading is counterintuitive. The reason for the vertical effective stress to increase is that the loading caused by the reservoir filling is higher than the consequent pore pressure increase caused by the undrained response of the low-permeability rock to compaction. We have modified a bit this sentence in an attempt to highlight this fact.*

50: Great introduction

*Author's response: We thank the reviewer for these positive assessment.*

52: I'm confused by the variation in abbreviations: Mw, ~M and M. All the same?

*Author's response: We use abbreviation "M" to refer to the magnitude of seismic occurrences; they most likely did not use the magnitude of the moment Mw. Nowadays, moment magnitude is the best representation of earthquake energy, but it cannot be used for all occurrences, hence we use M for all other types, which often employ body wave magnitude mb. As per very first RTS recorded, Lake Mead at the Hoover dam (United States) in the mid-1930s is associated with an approximate magnitude, we have used the symbol "~", i.e., "(~M) 4.0" to indicate that the value of the magnitude is approximately this value. This estimation is based on historical records, which showing some points of ambiguity regarding the precise magnitude of the events, as mentioned by Carder (1945). Apart from this, the subscript w is used when the magnitude refers to the local magnitude.*

59: Great

65: Thanks for the examples

68: What is the aim of the manuscript? It reads as if it was the forecasting of RTS hazard by interaction and analysis of drained and undrained responses.

*Author's response: We have modified this sentence to highlight that the aim is to understand the causes of RTS. Now, it reads "The interactions and comprehensive analysis of these two responses are key to understand the causes of RTS cases and eventually improve the forecasting and mitigation of RTS hazard".*

77: Great reading flow

*Author's response: We thank the reviewer for these positive words.*

85: Great position of this paragraph

93: that = it

*Author's response: We have corrected this in the revised manuscript.*

92: that = the

*Author's response: We have corrected this in the revised manuscript.*

96: I really like the traditional structure of the manuscript

106: Language issues? Word order?

*Author's response: Word order and correction have been made in the revised manuscript.*

104: Reference?

*Author's response: We have inserted the reference in the revised manuscript.*

105: ..sedimentation processes of passive margins associated with the deposition.. Language issues? Stands out against all other sentences.

*Author's response: We have rephrased this sentence.*

108: Blue colour doesn't match well in image/legend

*Author's response: We have made changes to the legend (blue colour) to match that of the image and have replaced the figure in the revised manuscript.*

108: What indicates the line at the top left?

*Author's response: The top left line and the one in the middle of the Chapada Acauã formation indicate a fault or compressed shear zone, as explained in the legend.*

114: Number = word?

*Author's response: We have changed this in the revised manuscript.*

124: Maybe word it more carefully? "This leads us to investigate a causative relationship/ hypothesis"

*Author's response: We have incorporated this suggestion.*

132: Space in reference missing

*Author's response: We have corrected this typo.*

147: Is this also true for the depth of the event (3.88km)? Maybe add information on faulting regime data source and depth.

*Author's response: Yes, it is true, Chimpliganond et al., 2007 assume that the stress regime does not vary within the first kilometres. Unfortunately, focal mechanisms could not be inferred from the seismic data, as explained in the text in the following sentence, which makes it impossible to assess the stress regime at depth.*

155: Why is this differently spelled? km vs. -km. It occurs several times throughout the manuscript.

*Author's response: When there are three nouns, the first two are hyphenized. For example, we write "long-term storage". For the case of "4.8-km thick" we could alternatively write "a thickness of 4.8*

*km". When km is followed by a noun, a hyphen should be included between the number and the units. This is why we write "4.8-km thick".*

182: Maybe stay in present tense here, like everywhere else.

*Author's response: Corrected.*

188: Delete second "an"

*Author's response: Deleted.*

220: ..in these samples. I find this btw already an interesting result!

*Author's response: It is true that there are several correlations to estimate permeability based on porosity. Nonetheless, permeability depends on the pore connectivity rather than the total volume of the pores. This is a good example that highlights this physical process.*

233: Maybe elaborate a little on the similarity of the rocks to make this argument more convincing for the reader.

*Author's response: We have mentioned a common similarity in the mineralogy between these two rock types to make the argument more convincing.*

276: I don't associate highly metamorphosed rock with good porosity. It reads as if the authors do. Maybe contrast better: "despite high porosity, the rock has low permeability. Therefore, …"

*Author's response: We thank the reviewer for this suggestion. We have incorporated it in the revised manuscript.*

278: I like the idea with the integrated outcrop picture. However, I don't find the figure intuitive to an amateur reader: Is this is a side view or a top view? What does the irregular trapezium indicate? Are the angles of the sides relevant? What do the yellow circles indicate? Where do the arrows point? Where does the crystalline rock start? Why is the figure in this irregular box?

*Author's response: We thank the reviewer for this idea. The diagram shows a vertical cross-section of the rock, with the reservoir in perspective. The irregular trapezium at depth illustrates how fault slips and RTS occurs due to the weight of the reservoir water volume. There is no relation to side angles; it is shown schematically. The yellow circles indicate the main earthquake. According to the local velocity model, the crystalline rocks start at a depth of 4.2 km. We have modified this figure to make it more understandable.*

290: What is the problem? Please specify.

*Author's response: The problem is that knowledge of the physics of the triggering mechanisms is required. To this end, the properties of the rock should be known. We now specify this in the revised manuscript.*

*Changes in the manuscript:*

*Page 21, lines 300-302: We added this line " Mitigation of the risk of RTS requires knowledge of the physical mechanisms that may trigger seismicity. Thus, a thorough characterization of the site to measure rock physical properties is crucial."*

289: "Potential reservoir site". Make it more obviously a recommendation and summary.

289 cont.: These are conclusions and recommendations rather than discussion. Move to correct section please.

*Author's response: We have clarified this recommendation and moved it to the conclusion in the revised manuscript.*

296: In the Irapé case the dam is already built and poro-perm measurements have been taken. Please make the distinction clearer between the existing project and potential future projects.

*Author's response: Yes, the dam is already built, but the porosity-permeability measurements have not been made until several years later after the filling of the reservoir. To the best of our knowledge, this is the first time that poro-perm measurements are made. We have made this distinction clearer in the revised manuscript.*

*Changes in the manuscript:*

*Clarifications have been made in the manuscript:*

*Page 21, lines 309-311: We added this line "Note that at Irapé, the porosity and permeability measurements have not been done until now, but should have been done prior to the design of the dam."*

299-309: PARAGRAPH IS POORLY WRITTEN, PLEASE REWRITE. Vocabulary is insufficient and words seem to be missing. While the rest of the manuscript is carefully and beautifully composed, this paragraph seems to lack structure, aim and, frankly, language skills.

*Author's response: We re-rewritten the paragraph.*

299: Is being employed or should be employed? Above, conditional tense is used, here not. Please adjust.

302: "effort have begun"? Maybe: Efforts have been made regarding... "infrastructure harm"? Maybe: damages… "loss and damages" of what?

300-302: Please rewrite. TLP does not initiate light and is not an approach or point. Maybe: According to the TLP green light permits operations without restriction, yellow light demands mitigation measures and red light appeals for regulatory intervention.

304: ensure??

305: can be revolved around??

307: promising what?

*Author's response: We thank to the reviewer for these recommendations to improve the text. We have re-written the paragraph in the revised manuscript.*

*Changes in the manuscript:*

*Page 21, lines 313-334: We have deleted the paragraph " Finally, to address and manage RTS risks, the traffic light protocol (TLP) is being employed. In general, TLP initiates the green light as the primary approach allowing operations without restrictions, the yellow light as the point to activate mitigation measures, and the red light as the point necessitating regulatory intervention. The efforts have also begun by linking the configuration of TLPs with risk-oriented measures, infrastructure harm, and the likelihood of loss or damages while adapting them to real-time data. The occurrences that may ensure after an operation are crucial, given their substantial impact on standard risk management. Nevertheless, these methodologies can be revolved around by assessment of events succeeding in the conclusion of an operation. The utilization of physics-based models holds promising by illustrating and projecting anticipated seismic activity, enabling the anticipation of future warnings and risks, and build up the information for operational adjustments and for future mitigation (Boyet et al., 2023b) (Figure 8).*

*We have added "To address and manage RTS risks, the Traffic Light Protocol (TLP) should be employed (Figure 8). A TLP is a tool that assists decision makers to decide how to operate the dam to minimize risks. The TLP has three levels of operation: (1) a green light that allows operations to proceed without restrictions, (2) a yellow light that requires to activate mitigation measures, and (3) a red light that urges to stop operation  Efforts have been made regarding the incorporation of real-time data with the application risk-oriented measures to prevent infrastructure damage and nuisance to the local community. Incorporating in TLP the two types of RTS, i.e., immediate events induced by the undrained response of the subsurface to water-level changes, and delayed seismicity induced by pore pressure diffusion, is crucial. To this end, the utilization of physics-based models is promising since they are capable of anticipating seismic activity, enabling operational adjustments for future mitigation of RTS risk (Boyet et al., 2023b) (Figure 8)."*

311: The figure is insufficiently included and introduced in the paragraph.

*Author's response: We have re-written the paragraph in the  revised manuscript and better introduce the figure now.*

313-318: The description is great.

*Author's response: We thank the reviewer for these positive assessment.*

335: Manipulation of reservoir loading refers to the impoundment. Mind that the study also suggests mitigation and management strategies for risk of RTS (289-323) which is not mentioned here. Add?

*Author's response: We have rewritten the last sentence of the Conclusions in the revised manuscript.*

359-513: LIST OF REFERENCES IS INCONSISTENT AND CONTAINS ERRORS, PLEASE REWRITE. - points missing - spaces missing - pages differently abbreviated - DOIs differently inserted - links inconsistently used - some spelling errors

*Author's response: We have corrected and improved the errors in the revised manuscript.*

**Response to comment by Nikita Bondarenko**

The article presents an analytical assessment of the undrained response impact on the state of stress change during reservoir impoundment and associated reservoir triggered seismicity. Authors present laboratory measured permeability and porosity to support the assumption of undrained response. The assessment is based on assumption of oedometer conditions and using analytical expression based on poroelastic relationships. However, a significant deficiency is that Equation 1 has only relation between horizontal stress and pore pressure. Even if the pore pressure is not changing (drained condition) the horizontal stress would change due to the change of vertical stress. This effect is expected to be the same order of magnitude or stronger than the effect of pore pressure change and therefore cannot be neglected. It is recommended that the calculations have to be updated or it should be explicitly shown that the term can be neglected (more details are available in specific comments in the attached file.

*Author's response: We extend our thanks to the reviewer for his careful consideration and positive assessment of the paper. We have addressed the reviewer's comments, which helped clarify some of our assumptions in order to improve the manuscript's quality. In particular, we have updated our calculations.*

Specific comments:

12: Either "Specifically" or "in particular"

*Author's response: We have deleted "in particular".*

22: Is the accuracy of porosimetry sufficient to report it to third decimal places?

*Author's response: According to the specifications of the apparatus, it is sufficient.*

69: The delayed response is dominant in case of the Koyna, however the local maximum of seismic activity is also observed during the maximum water level (http://dx.doi.org/10.1134/S1069351322030077). It might be worthy to mention that both mechanisms are present, but the dominant one is delayed.

*Author's response: We now mention both mechanisms in Table 1.*

122-123: It would be great to mention the depth of earthquakes here.

*Author's response: We now mention the depth of the largest earthquake in the main text and refer to Table S1 for the depth of the other earthquakes.*

*Changes in the manuscript:*

*Page 8, lines 119-120: We added the sentence "The epicenters are distributed from 0 to 11.4-km depth, showing a progressive increase in depth (see Table S1)."*

127: The number of events cannot be negative, so it would be better to start the vertical axis from 0 and not -1.

*Author's response: We have corrected the vertical axis to start at 0 in Figure 2 in the revised manuscript.*

134: Since only three stations were utilized for the location, it seems to be impossible to determine 4 parameters of earthquake hypocenter (x,y,z,t) without certain assumptions. It would be great to mention if assumptions were made and what is average accuracy of location.

*Author's response: We thank the reviewer for this comment. The authors concur with the reviewer's perspective and have added some additional explanations in the revised manuscript. The positional uncertainty of seismic events is influenced by the quality and spatial distribution of seismometers in the region. At Irapé, the uncertainty is notably high due to operational challenges encountered by the local monitoring stations.*

*Changes in the manuscript:*

*Clarifications have been made in the manuscript:*

*Page 10, lines 135-139: We added the sentence "The local monitoring station presented operational challenges, which resulted in positional uncertainty of seismic events (Silva et al., 2014). The velocity model that was used to locate the seismic events was based on a deep seismic refraction survey in combination with local geological interpretations and studies of the crustal structure in south-eastern Brazil (Assumpçao et al., 2002b)."*

184-186: It would be great to mention the lower instrumental limit for the permeability and the accuracy of measurements. It also seems to be inconsistent with the information in supplementary material, which says that 0.01 mD is the lower limit. In general, supplementary material contains important information but never referred in the main manuscript which make it difficult to find.

*Author's response: We thank the reviewer for this comment. The authors concur with the reviewer's perspective on the accuracy of the measurements. The lower instrumental limit for the permeability is 0.002 mD, as we now mention in the Supplementary Material (we had a typo in the previous version). We obtained data from most of the measured samples, which present permeability below the precision of the apparatus.*

192 MAJOR: The equation consider only relationship between change in horizontal stress and change inpore pressure. However, the change in vertical stress would also cause change in the horizontal stress, which is neglected. The equation should be:

$$\Delta\sigma_h = \frac{\nu}{(1-\nu)}(\Delta\sigma_v) + \alpha\frac{(1-2\nu)}{(1-\nu)}\Delta p \qquad (1)$$

*Author's response: We thank the reviewer for this suggestion. The authors agree with this added term, as the increase of the vertical stress induces an increase in the horizontal stress. We have added this equation in the revised manuscript.*

197: What is the sign agreement here? If the pore pressure is positive and compressive stresses are also positive, it should be no minus sign.

*Author's response: We have corrected this typo.*

198: "mean stress change"

*Author's response: We have added this word.*

201: Equation seems to assume that there is no minus sign in Equation 2. MAJOR: The equation have to be modified due to the change in Equation 1. Or it should be explicitly shown why term proportional to change of vertical stress in neglected.

*Author's response: Yes, we assumed the sign criterion of geomechanics in Equation (2). We have updated Equation (3) to take into account the change in Equation (1).*

*Changes in the manuscript:*

*Clarifications have been made in the manuscript:*

*Page17, lines 238-241: "Pore pressure increased by 0.61 MPa in response to an increase of 136 m in the reservoir-water level. The vertical effective stress increased by 0.75 MPa and the horizontal effective stress decreased by 0.48 MPa."*

224: Is the porosimetry is accurate to report it to three decimal places? It is suggested that "<0.002" is reported instead of "0.002" for permeability.

*Author's response: We have modified Tables 3 and S2.*

226: Since the measurement are close to instrumental limit, it would be great to show this limit on the plot as well as the accuracy of calculated permeability

*Author's response: We have shown in revised manuscript.*

242: It is mentioned further in the text, but it would be great place to say that absence of fractures is assumed.

*Author's response: Thanks for the suggestion, we added this text in the revised manuscript.*

259-261: The permeability measured in the lab is usually 1-2 orders of magnitude lower than the in-situ one even without visible fractures. At least in the laboratory experiments, the presence of open fractures is capable of enhancing the permeability by 5 orders of magnitude (e.g., https://doi.org/10.1038/s41598-022-19775-4 or https://doi.org/10.22541/essoar.171629597.77744897/v1). It is acceptable to assume that 5 orders of magnitude enhancement is unlikely, but slightly extended discussion would be worthy here.

*Author's response: We thank to reviewer for this insightful suggestions. We have extended this discussion in the revised manuscript.*

*Changes in the manuscript:*

*Page17, lines 263-268: Extended discussion have been added in the manuscript: "The permeability enhancement due to the presence of fractures could become larger in crystalline than in clay-rich rock, reaching an increase of up to five orders of magnitude (Bondarenko et al., 2022). Such high permeability enhancement caused by fractures is not feasible in clay-rich rock like mica-schist because of its ductility and low dilatancy angle, which prevents fractures from becoming open pathways. At Irapé, the necessary permeability of the rock to reach the depth of the largest earthquake within 0.5 years".*

334-335: It is not fully clear what is meant by "carefully manipulating reservoir loading". If the change of impoundment rate is meant, it seems unlikely that the impact of undrained response will be different by the end of impoundment. If the limiting the maximum water level is meant, it is not fully clear what is suggested as a limit and rewording might be necessary.

***Author's response:*** *We have re-written the last sentence of the Conclusions. We hope it is clearer now.*